# Shedding Light on the Drug–Target Prediction of the Anti-Inflammatory Peptide *Tn*P with Bioinformatics Tools

**DOI:** 10.3390/ph15080994

**Published:** 2022-08-12

**Authors:** Carla Lima, Silas Fernandes Eto, Monica Lopes-Ferreira

**Affiliations:** 1Immunoregulation Unit of the Laboratory of Applied Toxinology (CeTICs/FAPESP), Butantan Institute, Sao Paulo 05503-900, Brazil; 2Industrial Development and Innovation Laboratory, Butantan Institute, Sao Paulo 05503-900, Brazil

**Keywords:** cyclic peptides, *Tn*P, bioinformatics and computational tools, drug–target protein interactions, in silico prediction, integrin antagonist

## Abstract

Peptide–protein interactions are involved in various fundamental cellular functions, and their identification is crucial for designing efficacious peptide therapeutics. Drug–target interactions can be inferred by in silico prediction using bioinformatics and computational tools. We patented the *Tn*P family of synthetic cyclic peptides, which is in the preclinical stage of developmental studies for chronic inflammatory diseases such as multiple sclerosis. In an experimental autoimmune enceph-alomyelitis model, we found that *Tn*P controls neuroinflammation and prevents demyelination due to its capacity to cross the blood–brain barrier and to act in the central nervous system blocking the migration of inflammatory cells responsible for neuronal degeneration. Therefore, the identification of potential targets for *Tn*P is the objective of this research. In this study, we used bioinformatics and computational approaches, as well as bioactivity databases, to evaluate *Tn*P–target prediction for proteins that were not experimentally tested, specifically predicting the 3D structure of *Tn*P and its biochemical characteristics, *Tn*P–target protein binding and docking properties, and dynamics of *Tn*P competition for the protein/receptor complex interaction, construction of a network of con-nectivity and interactions between molecules as a result of *Tn*P blockade, and analysis of similarities with bioactive molecules. Based on our results, integrins were identified as important key proteins and considered responsible to regulate *Tn*P-governed pharmacological effects. This comprehensive in silico study will help to understand how *Tn*P induces its anti-inflammatory effects and will also facilitate the identification of possible side effects, as it shows its link with multiple biologically important targets in humans.

## 1. Introduction

The *Tn*P family invention, currently patented (Appendix A) in several countries, refers to synthetic cyclic peptides found in the venom of the Brazilian fish *Thalassophryne nattereri*. *Tn*P, in a preclinical development stage, is known for its therapeutic potential in chronic inflammatory diseases such as multiple sclerosis (MS) [1]. Proof of concept supporting its anti-inflammatory effect was found using the animal model of experimental autoimmune encephalomyelitis (EAE). The valuable potential of the patented *Tn*P family for controlling neuroinflammation and preventing demyelination is due to its systemic ability to interfere in the dynamic circuit of immune cell groups, as well as locally in the central nervous system (CNS).

We found that subcutaneous treatment with *Tn*P successfully improves the severity of clinical signs of myelin oligodendrocyte glycoprotein (MOG)-induced EAE, slowing down by 4 days the onset of maximal symptoms and decreasing by 40% the severity of symptoms, compared with control EAE mice treated with vehicle alone. *Tn*P amends EAE in an IL-10-dependent way, including suppressing activation of conventional dendritic cells (DC) and providing the emergence of plasmacytoid DC and regulatory cells during the EAE induction phase; blocking the transit and infiltration of leukocytes into the CNS by suppressing matrix metalloproteinase (MMP)-9 activity and CD18 expression; blocking the reactivation and permanence of Th (T helper)1 and Th17 lymphocytes in the CNS; preventing of microglial expansion and macrophage infiltration into the CNS; favoring the localized increase in regulatory T cells; and finally, suppressing demyelination in the spinal cord of EAE mice leading to accelerated remyelination dependent on blocking leukocyte migration in a cuprizone model [2].

The evidence that *Tn*P can be efficient in crossing the blood–brain barrier (BBB), which was described in zebrafish larvae [3], and can act locally in the CNS blocking the migration of inflammatory cells responsible for neuronal degeneration are features that lead us to hypothesize a potential ability of *Tn*P to interact with membrane receptors that control leukocyte traffic to the CNS. Therefore, the identification of potential targets for *Tn*P is fundamental at this stage of preclinical studies.

Drug–target interactions play key roles in drug discovery and development. The identification of drug targets is useful to know the drug’s mechanism of action but also to identify possible adverse side effects and plan its repositioning. The drug–target interactions can be inferred by in silico prediction of interactions between drugs and target proteins using bioinformatics and computational tools [4].

In this study, we used bioinformatics and computational approaches, as well as bioactivity databases, to evaluate *Tn*P–target prediction for proteins that were not experimentally tested, specifically predicting the 3D structure of *Tn*P and its biochemical characteristics, *Tn*P–target protein binding and docking properties, and dynamics of *Tn*P competition for the protein/receptor complex interaction, construction of a network of connectivity and interactions between molecules as a result of *Tn*P blockade, and analysis of similarity with bioactive molecules. Our results showed promising molecular targets of *Tn*P, including integrins with pharmacological action on leukocyte migration. The in silico model built to predict the possible molecular targets of *Tn*P showed a harmonic resonance between the anti-inflammatory effects and computational modeling.

## 2. Results and Discussion

### 2.1. Bioinformatics and Computational Tools Predict Oral Bioavailability of *Tn*P

More recently, cyclic peptides have gained significant attention for use as candidate therapeutics due to the high specificity and high affinities they can achieve against a wide range of targets [5]. The anti-inflammatory activity of *Tn*P was elucidated in EAE [2], and its treatment efficacy was recently compared with current disease-modifying therapies [6], but the molecular targets of interest are still unknown and remain highly challenging.

This study was conducted with the aim to extend the knowledge of the mode of action of *Tn*P from the understanding of its interaction with protein targets. To start the *Tn*P–target prediction search, we redrew its primary linear structure (Ile–Pro–Arg–Cys–Arg–Lys–Met–Pro–Gly–Val–Lys–Met–Cys) with 13 L-amino acids using the simplified molecular-input line-entry system (SMILES) code according to Weininger [7] to a 2D chemical structure (Figure 1A). The Protein Data Bank (PDB) structure showed a bridge between Cys-4 and Cys-13 that gives *Tn*P a clip-like shape; the amino acids Ile-1, Pro-2, and Val-3 point toward the outside of the circular structure, which may contribute to van der Waals and hydrogen bonding interactions when binding to target (Figure 1B).

Compared with linear peptides, *Tn*P is stable and less prone to protease degradation such as trypsin and pepsin [1]. The low toxicity found in preclinical toxicology studies in zebrafish suggests transdermal absorption and a rapid in vivo clearance [3], which could indicate the need for a large number of such peptides. Furthermore, during the pharmacotechnical phase, the use of pharmacological and biotechnological alternatives, such as the design of delivery-directed systems, can be applied to improve the *Tn*P capacity for longer blood circulation [8].

Bioinformatics and computational approaches have been fostered as valid alternatives to experimental procedures for the prediction of absorption, distribution, metabolism, and excretion (ADME) parameters, especially during the initial stages of drug development [9]. Figure 1C illustrates the radar of oral bioavailability of *Tn*P, where the pink-colored zone indicates the ideal physicochemical properties for the compound to perform a pharmacological activity in the human organism, while the red indicator marks the specific index of each property for the tested compound. We observed that red indicators of flexibility and polarity show *Tn*P as too flexible and polar with a high number of rotatable bonds and a topological polar surface area (TPSA) higher than 130 Å^2^. The predicted value of the size (higher than 500 g/mL) corresponds to the actual molecular weight (MW) of 1514.8 Da. In contrast, the indicators of lipophilicity, solubility, and saturation for *Tn*P are in the pink zone, indicative of oral bioavailability. We showed earlier that the high solubility of *Tn*P allows it to be quickly absorbed by the subcutaneous route of administration [2,6].

Longer cyclic peptides can form secondary structures upon binding, offering low nanomolar values of binding interactions (hydrophobic, van der Waals, polar, and hydrogen bonding) that are able to access intracellular proteins involved in disease complexes. McAllister et al. [10] analyzed cyclic peptides obtained by various technologies against a protein target (phage-display, mRNA-display, and split-intein circular ligation of peptides and proteins—SICLOPPS) between 2015 and 2019 in more than 40 publications, and interestingly, the MWs of the identified cyclic peptides cover a relatively wide range, from 443 to 2717 Da.

Figure 1D shows the prediction of BBB penetration (yellow area) and gastrointestinal absorption (white area) of *Tn*P using the BOILED-EGG method based on Log P and TPSA values. The analysis of *Tn*P pharmacokinetics revealed that neither the white area of gastrointestinal absorption nor the yellow area of BBB penetration was found in the gray zone, indicative of high gastrointestinal absorption and permeability through the BBB, which corroborates our in vivo data [2,3,6,11]. An effective neurologic drug should be able to permeate the BBB so as to bind to specific receptors and initiate signaling pathways [12].

### 2.2. In Silico *Tn*P–Target Protein Interactions

We searched for possible protein targets of *Tn*P using the SwissTargetPrediction, a web interface able to predict the targets of bioactive small molecules by combining 2D and 3D similarity measures [13]. There are many types of drug targets, such as G protein-coupled receptors (GPCRs), protein kinases, enzymes, ion channels, and transporters [14].

In Figure 2A, we observe that the most prevalent top targets for *Tn*P interaction are enzymes as Eraser-associated proteins, which remove specific post-translational modifications (PTMs) from histone substrates with 32% and kinases (8%), and membrane receptor proteins (24%) and Family A of the GPCRs or rhodopsin family (20%). Enzymes account for 40% of the preferred targets of *Tn*P, while membrane receptors together represent 44% of predicted targets. These data support the finding that, beyond its capacity to inhibit serine proteases [1], *Tn*P can offer other intrinsic properties that contribute to the termination of the inflammatory process, including modulation of integrin expression, signal transduction, and tissue remodeling.

Then, we tried to identify the high-affinity bindings that occur between *Tn*P and the pockets of these target proteins in static modeled 3D structures [15,16,17] by calculating the free energy of the binding complex score, which is often used to determine the affinity of biomolecular interactions and the efficacy of drugs. Figure 2B shows highly negative scores of free energy of binding, ranging from −215 to −200 kcal·mol^−1^, which signifies very strong binding to all targets, including integrins of beta subunit such as ITB1 and ITB7 or alpha subunit as alpha 4 (ITGA4 or ITA4) and alpha-IIb (ITA2B); furin, a type 1 membrane-bound protease member of the subtilisin-like proprotein convertase family; and two members of melanocortin receptor family (MCR)-4 and MCR-3), with central roles in weight regulation. Interestingly, we recently demonstrated that the improvement in the clinical score induced by *Tn*P treatment in EAE mice corresponded to an increase in body weight immediately after disease induction, consistent with its role in controlling weight regulation [6].

The root-mean-square deviation (RMSD) was performed to find the simulation result stabilities confirmed low scores within an acceptable range for all *Tn*P targets and indicated stable complex formation between *Tn*P and ITB1, ITB7, ITA4, ITA2B, and furin (Figure 2C). It is very interesting to mention that the ITA4 (alpha 4, CD49d) and ITB1 (beta 1, CD29) subunits form the integrin α4β1 dimer known as very late antigen-4 (VLA-4) with a broad expression on immune cells.

According to the predicted free energies of binding, showing the integrin α4β1 as one of the most probable targets of *Tn*P, as shown in Figure 2D, we evaluated the binding poses of *Tn*P within the ITA4 binding site, as well as their overlapping structures. *Tn*P was well-embedded in the active pocket of the integrin subunit, as observed from different angles.

The analysis of the contribution of individual residues for high-affinity bindings using the 3D structure of human ITA4, available from the α_4_β_7_ integrin complex (PDB code: 3V4V) was obtained, according to Yu et al. [18]. The molecular docking study showed that the hydrophobic interactions with amino acids residues present in the ITA4 binding site such as Phe-191, Phe-453, Pro-216, Ile-202, Trp-126, Trp-221, and Met-429, as well as Gln-274, Gln-392, Gln-199, Ser-309, Tyr-250, and His-146, were among the highest contributors to the total binding energy. As demonstrated, the van der Waals energy, electrostatic energy, and polar solvation energy formed from the interactions of these amino acids with *Tn*P might have contributed to the highly negative total binding energies and, consequently, promoted the stabilization of the complex [19,20].

### 2.3. *Tn*P Competition for the ITA4/VCAM-1 Complex Interaction

The two solved N-terminal immunoglobulin (Ig) domains of human vascular cell adhesion protein (VCAM-1) revealed that both domains adopt a β–β sandwich topology, composed of an anti-parallel array of β-strands. As predicted by homology modeling, the IDSP motif (Ile–Asp–Ser–Pro) located on a projected loop between β-strands C and D, mainly the aspartate residue, is critical to the active sites of the integrin ligand [21,22,23]. VLA-4 binding sites in domains 1 and 4 minimally involve amino acids within a linear sequence of six amino acids that are identical in both domains: Gln-38–Ile-39–Asp-40–Ser-41–Pro-42–Leu-43 in domain 1 and Gln-326–Ile-327–Asp-328–Ser-329–Pro-330–Leu-331 in domain 4 [24].

Then, we analyzed the competitive *Tn*P–target interaction dynamics based on the affinity similarity between ITA4 and VCAM-1 interaction using a computational prediction of drug–target interactions (DTIs). The simulations were carried out using the Hdock Serve model, according to Gao and Skolnick [25]. The interaction/docking of the ITA4/*Tn*P/VCAM-1 complex was investigated, and the results showed a marked reduction in the free energy of the binding complex for the 10 simulations in relation to the interaction of the ITA4/VCAM-1 complex (Figure 3A, right), and the same reductive pattern was observed for individual interactions (Figure 3A, left). The spatial arrangement of the ITA4/*Tn*P/VCAM-1 complex showed that *Tn*P blocks (Figure 3B) were the main activity sites of the VCAM-1 pocket composed of the amino acids Thr-37, Gln-38, Ile-39, Asp-40, Ser-41, Pro-42, and Leu-43, detailed in a square of Figure 3C. This competitive interaction appeared sufficient to perturb ITA4/VCAM-1 interaction, inducing drastic structural changes at interacting protein surfaces.

### 2.4. Connectivity and Interactions of Molecules as a Result of *Tn*P Blockage

Networks have been built to represent possible applications of molecular interactions from *Tn*P–target [26,27]. To achieve this, STRING collection was limited to *Homo sapiens,* and scores were employed to construct a functional association network (Figure 4). According to the prediction, *Tn*P blockage of the ITA4/VCAM-1 binding generates compromise posterior signaling through metalloproteinases, such as MMP-9, MMP-3, and MMP-14, that directly degrade extracellular matrix (ECM) proteins and activate cytokines and chemokines to regulate tissue remodeling, as well as the tissue inhibitor of matrix metalloproteinase (TIMP)-1, which have been highlighted in blue. Adhesion molecules including other integrins (such as alpha 1, alpha 3, and alpha 6) and vinculin (VCL) that link the core proteins to the subcortical actin cytoskeleton, and PTK2B (protein tyrosine kinase 2 beta), highlighted in green, occur as a result of the interruption of ITA4/VCAM-1 complex interaction by *Tn*P.

In addition, integrin-mediated signaling molecules (regulator of kinases protein: CRK, a cytosolic tyrosine kinase that negatively regulates Src family kinases: CSK, adapter protein: BCAR1, mammalian isoforms of talin 1: TLN1, and PTK2B) that are involved in physiological processes, including regulation of immune systems, cellular motility, and differentiation, are highlighted in red. Proteins without highlights (white) entered the network based on close associations to the ITA4/*Tn*P/VCAM-1 complex.

Furthermore, the network interactions originated from the *Tn*P blocking of the ITA4/VCAM-1 interaction showed the formation of two clusters of fusion proteins: a cluster of two interactions included SRC (Src family protein tyrosine kinases member), FN1 (glycoprotein in the ECM Fibronectin), and integrins ITGAV, ITGA5, ITGA2, and MAdCAM-1 (addressin mucosal addressin cell adhesion molecule-1); and a cluster of three interactions that included ITGB3, ITGB1, ITGB7, and ITGA4 (ITA4). Both fusion clusters are largely associated with biological processes including the movement of inflammatory cells, locomotion, and localization of cells, cell motility, and cell migration [28].

However, whether *Tn*P disrupts the rolling, adhesion, spread, and migration of cultured cell lines to α4β1 ligands such as VCAM-1 under static and flow conditions is still an issue to be investigated in the future. In addition, valuable proof-of-concept data could be obtained from systemic analyses in animal models, such as zebrafish (*Danio rerio*) [29,30].

### 2.5. Similarity Analysis of *Tn*P with Molecules from the Swiss Similarity Database

Chemical similarity states that similar chemical structures more often than not tend to present similar biological activities, although even structurally similar compounds may interact with a protein target in different ways [31]. Next, we applied the SWISS-MODEL [32,33] homology modeling tool enhanced by ChEMBL databases [34] and ElectroShape virtual screening methods that combine shape and electrostatic information [35] to identify potential drug candidates that can interact with the protein of interest. Figure 5 shows the molecules/drugs that had the highest similarity scores, selected based on enrichment ratios of around 1%. We highlighted bioactive molecules that bind to the same biological target of *Tn*P—mainly CHEMBL414360 (MW: 1490.83, cytochrome p450 ligand); CHEMBL385688 (MW: 1390.61, probable G-protein coupled receptor 132 ligand); and Desmopressin (MW: 1069.22, acting as a selective agonist of V2 receptors). Together, our data showed that the identification of putative ligands of *Tn*P helps to explain its mechanisms of action and repositioning possibilities.

## 3. Methods

### 3.1. Virtual Construction of the *Tn*P Peptide

The 2D and 3D structures of the *Tn*P peptide were predicted and constructed virtually on the PEPstrMOD server (https://webs.iiitd.edu.in/raghava/pepstrmod/) (9 May 2022) by inserting the sequence of the 13 L-amino acids, a disulfide bridge, forming a cyclic peptide. The chemical structures were retrieved with individual database (IDs) and simplified molecular-input line-entry system (SMILES) code. The result generated a Protein Data Bank (PDB) file with the predicate structure of *Tn*P.

### 3.2. ADMET In Silico

SwissADME (http://www.swissadme.ch/ 9 May 2022) was used for the predictive study of the pharmacokinetic properties of *Tn*P. Simulations of ADME were performed. This test includes the analysis of absorption and permeability of the drug candidate by the partition coefficients (Log P) and topological polar surface area (TPSA) to evaluate the permeation capacity of biological membranes.

### 3.3. Estimation of Macromolecular Targets of *Tn*P with Homo Sapiens Proteins

The physicochemical properties, as well as the estimation of the possible molecular targets of the *Tn*P peptide isolated from the venom of *Thalassophryne nattereri,* were virtually predicted using the SwissTargetPrediction server (http://www.swisstargetprediction.ch/ 9 May 2022). For this, the following SMILES sequence of *Tn*P was entered into the server: CCC(C)C(N)C(=O)N1CCCC1C(=O)NC(CCCN=C(N)N)C(=O)NC2CSSCC(NC(=O)CCCSC)NC(=O)C(CCCCN)NC(=O)C(NC(=O)CNC(=O)C3CCCN3C(=O)C(CCSC)NC(=O)C(CCCCN)NC(=O)C(CCCNC(N)= N)NC2=O)C(C)C)C=O; the *Homo sapiens* protein database was selected as a data source for the analysis of possible interactions.

### 3.4. Computational Analysis of the Molecular Docking Interactions of *Tn*P with the Human ITA4 Molecule

We analyzed the possible pharmacological mechanism of *Tn*P through virtual docking interactions in three ways. First, on the Hdock Serve server (http://hdock.phys.hust.edu.cn/ 9 May 2022), we analyzed the interaction of *Tn*P with the alpha4 integrin subunit (ITGA4, which was predicted and built using SWISS-MODEL (https://swissmodel.expasy.org/ 9 May 2022) *H. sapiens* reference 4irz. In total, 100 models were generated, and only 10 models belonging to the top 10 were selected using the free binding value. After choosing the most representative model, the model (peptide–protein complex) was submitted to redocking using the flexpwpdock server (http://flexpepdock.furmanlab.cs.huji.ac.il/ 9 May 2022). In the second step, we analyzed the interaction of VCAM-1 with ITA4, and in the third, we compared this complex (ITA4/VCAM-1) with the complex (ITA4/*Tn*P/VCAM-1). Finally, we compared the values of free binding energy between the two complexes (with and without *Tn*P) scores. The stability of the docked complexes was analyzed using the root-mean-square deviation (RMSD) scores generated from the molecular dynamics (MD) simulations. Structures of protein targets were obtained from PDB (https://www.rcsb.org/ 9 May 2022).

### 3.5. Functional and Pathway Enrichment Analysis

The STRING database was used to perform functional enrichment analysis and to construct a protein–protein interaction (PPI) network, to subsequently identify *Tn*P-associated genes (https://string-db.org/cgi/ 9 May 2022).

### 3.6. Similarity of *Tn*P with Molecules from the Swiss Similarity Database

To analyze the similarity of *Tn*P with biomolecules or with existing chemotherapeutics, we used the SwissSimilarity server database (http://www.swisssimilarity.ch/ 9 May 2022). For that, we inserted the SMILES sequence of the *Tn*P in the server, and the scan was performed using the ElectroShape methods, spectrophores, and combined UYYOR–MYSXXWQL–UHFFFAOYSA-N method for bioactive databases, as well as the Food and Drug Administration (FDA) database. Other relevant information on ligands was collected from the PubChem (https://pubchem.ncbi.nlm.nih.gov; assessed on 25 December 2020) and CHEMBL (http://www.ebi.ac.uk/chembl 9 May 2022).

## 4. Conclusions

Our findings demonstrated a multiplicity of *Tn*P actions [11,36] that are in line with the new trend in the development of peptide drugs for the immunotherapy of immune-mediated chronic inflammatory diseases [37,38]. The ability of cyclic peptides to modulate the immune response is predicted to be associated with their simultaneous action on multiple molecular targets [39].

Among the multiple immunomodulatory effects of *Tn*P, it is interesting to note its action in the regulation of the entry of inflammatory cells into the CNS, inhibiting the expression of integrin receptors in the BBB that precedes perivascular infiltration and disease onset [2]. Targeting α4β1 integrin could be a potentially effective approach for the treatment of several diseases such as multiple sclerosis [40,41].

Integrin α4β1 is best known as a lymphocyte integrin that mediates the adhesion of circulating lymphocytes to VCAM-1, fibronectin, and osteopontin (OPN) expressed on activated endothelial cells, thereby promoting extravasation of lymphocytes into inflamed tissue [42]. However, some studies have suggested roles for integrin α4β1 and its ligand VCAM-1 in angiogenesis during blood vessel formation and particularly in inflammatory angiogenesis and neovascularization in vivo [43,44]. Besides α4β1 expression by tumor cells conferring them the ability of vascular adhesion and transmigration out of the blood flow, α4β1 expressed by myeloid cells contributes to tumorigenicity and metastasis in different manners [45,46].

Thus, it is evident that targeting α4β1 integrin can be valuable for the treatment of a variety of cancers that require a premetastatic niche in the development of metastases [47]. Selective integrin ligands have been widely used to target tumors with integrin overexpression, as inhibitors of cancer angiogenesis [48,49] and blockers of excessive blood clotting [50].

In this study, using bioinformatics tools, we investigated the possible molecular targets of *Tn*P that corroborate its action in the control of leukocyte migration into the CNS described in a murine model of multiple sclerosis. The in silico model built to predict the possible molecular targets of *Tn*P showed a harmonic resonance between the anti-inflammatory effects and the computational modeling. We found that the most stable conformations of *Tn*P with target proteins occurred mainly with integrins, such as ITB1, ITB7, ITA4, and ITA2B. Specifically, the mode of binding of *Tn*P to the target protein showed that it overlapped the ITA4/VCAM-1 binding cleft, leading to a decrease in complex interaction, with a subsequent reduction in the binding strength. As a result, integrin may remain in a bending form, which decreases its activation (Figure 6).

Activation of the ITA4 subunit of VLA-4 induced in an inflammatory context is very rapid and generates changes in its surface density with an increase in affinity and avidity [51]. Therefore, competition of the integrin-receptor binding cleft via *Tn*P could, at a biochemical level, alter the overall binding potential, compromising dissociation kinetics and thermodynamic equilibrium [52] with a consequent decrease in cell adhesion.

Interestingly, our similarity analysis data suggested that *Tn*P may cross-react with other integrins such as α4β7 (implicated in homing to Peyer’s Patches, a receptor for MAdCAM-1) as well as αIIBβ3 (a highly abundant heterodimeric platelet receptor playing a role critical in hemostasis and thrombosis, necessary for platelet aggregation). Such structural studies offer an opportunity for *Tn*P as a novel peptide drug candidate for therapy for other extremely promising therapeutic areas, from autoimmunity to thrombotic vascular diseases and cancer metastases.

In conclusion, we demonstrated that bioinformatics and computational analysis for peptide–protein interaction prediction help to clarify the identification of *Tn*P–target interactions, but also facilitate the identification of possible side effects, as they reveal the links with multiple biologically important targets in humans.

## Figures and Tables

**Figure 1 pharmaceuticals-15-00994-f001:**
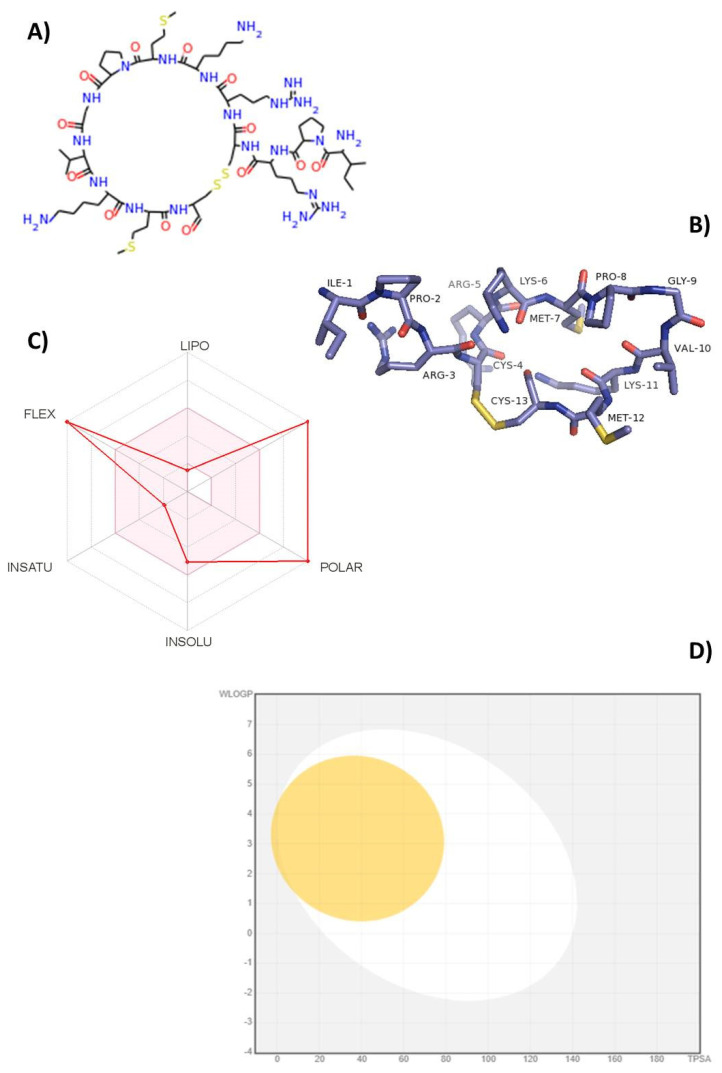
Two-dimensional chemical structure and bioavailability of *Tn*P: (**A**) two-dimensional chemical structure of *Tn*P was predicated by SMILES sequence; (**B**) three-dimensional structure of *Tn*P composed of 13 amino acids and a cyclic disulfide bond predicated by SwissTargetPrediction; (**C**) bioavailability radar chart for *Tn*P. The pink zone represents the physicochemical space for oral bioavailability, and the red line represents the oral bioavailability properties; (**D**) predicted BOILED-Egg plot from Swiss ADME online web tool for *Tn*P. The white region is for high probability of passive absorption by the gastrointestinal tract, and the yellow region (yolk) is for high probability of brain penetration.

**Figure 2 pharmaceuticals-15-00994-f002:**
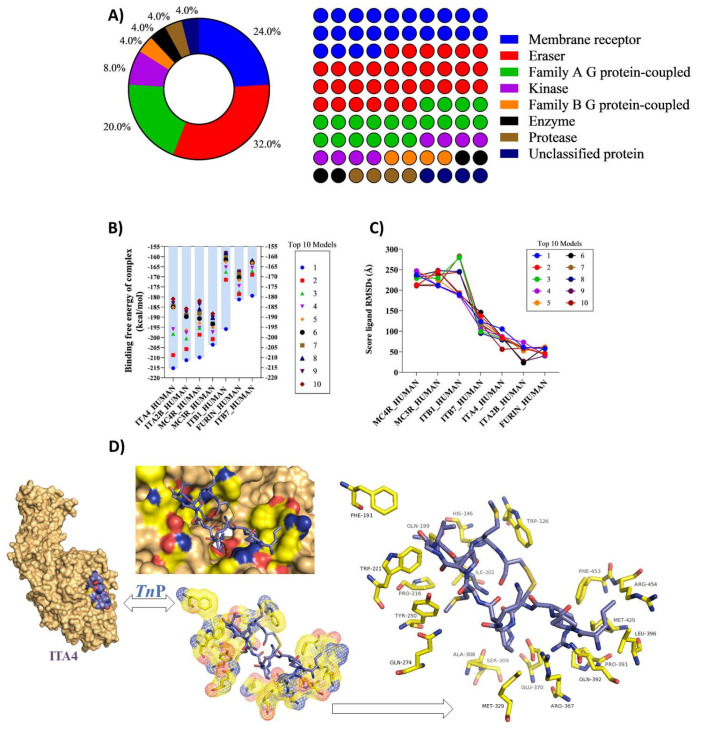
In silico *Tn*P–target protein interaction: (**A**) percentage and heat plot showing the main groups of human proteins/enzymes that interact with the *Tn*P predicate in SwissTargetPrediction; (**B**) free Energy of binding complex expressed in kcal/mol values of ten models simulated on the Hdock Serve platform (http://hdock.phys.hust.edu.cn/ accessed on 9 May 2022); (**C**) RMSD ligand score showing values in angstrom (Å) of the ten main models of simulated interaction with human molecules, and (**D**) the 3D structure of the *Tn*P/ITA4 complex and main ligand amino acids present in ITA4. The ITA4 PDB file was acquired from SWISS-MODEL (https://swissmodel.expasy.org/repository/uniprot/Q3MU74 accessed on 9 May 2022) using the ITA4_HUMAN as reference.

**Figure 3 pharmaceuticals-15-00994-f003:**
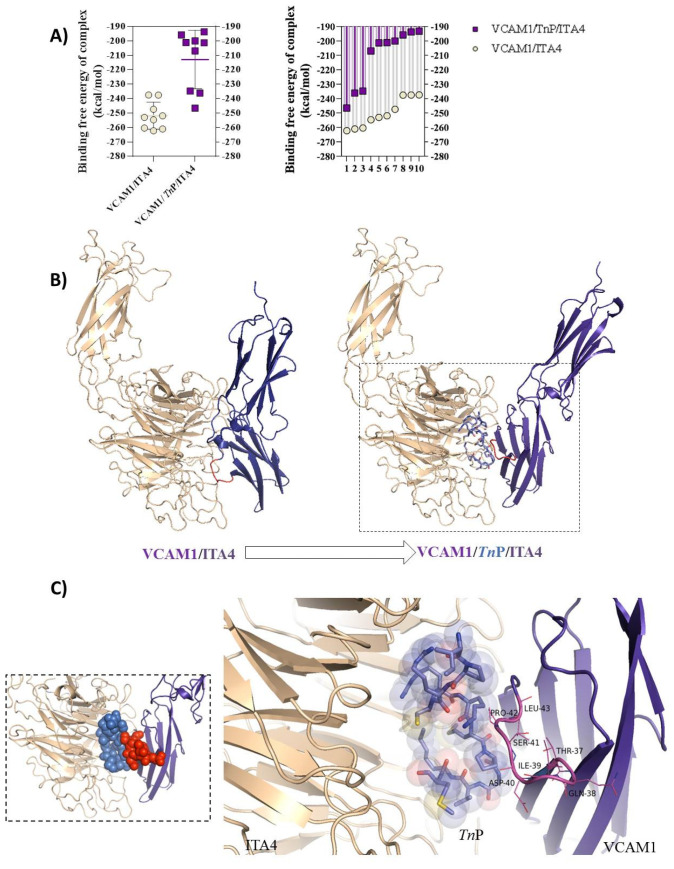
Model of the ITA4/VCAM-1 and ITA4/*Tn*P/VCAM-1 complex interactions: (**A**) docking showing the free energy of binding complex expressed in kcal/mol values of 10 models simulated on the Hdock Serve platform (http://hdock.phys.hust.edu.cn/ accessed on 9 May 2022) of the ITA4/VCAM-1 complex and ITA4/*Tn*P/VCAM-1 (on the **left**), and on the **right**, we observe the 10 main interactions of each complex analyzed separately; (**B**) a 3D cartoon structure of the ITA4/*Tn*P/VCAM-1 complex showing the main amino acids in the ITA4/VCAM-1 interaction responsible for binding to the ITA4 activity site in red, and in the square, we observe the *Tn*P blockade in the ITA4/VCAM-1 interaction; (**C**) details of the ITA4/*Tn*P/VCAM-1 complex and essential amino acids for active binding of the ITA4/VCAM-1 complex. The PDB file ITA4 and VCAM-1 were acquired from SWISS-MODEL using the reference ITA4_HUMAN and VCAM-1_HUMAN.

**Figure 4 pharmaceuticals-15-00994-f004:**
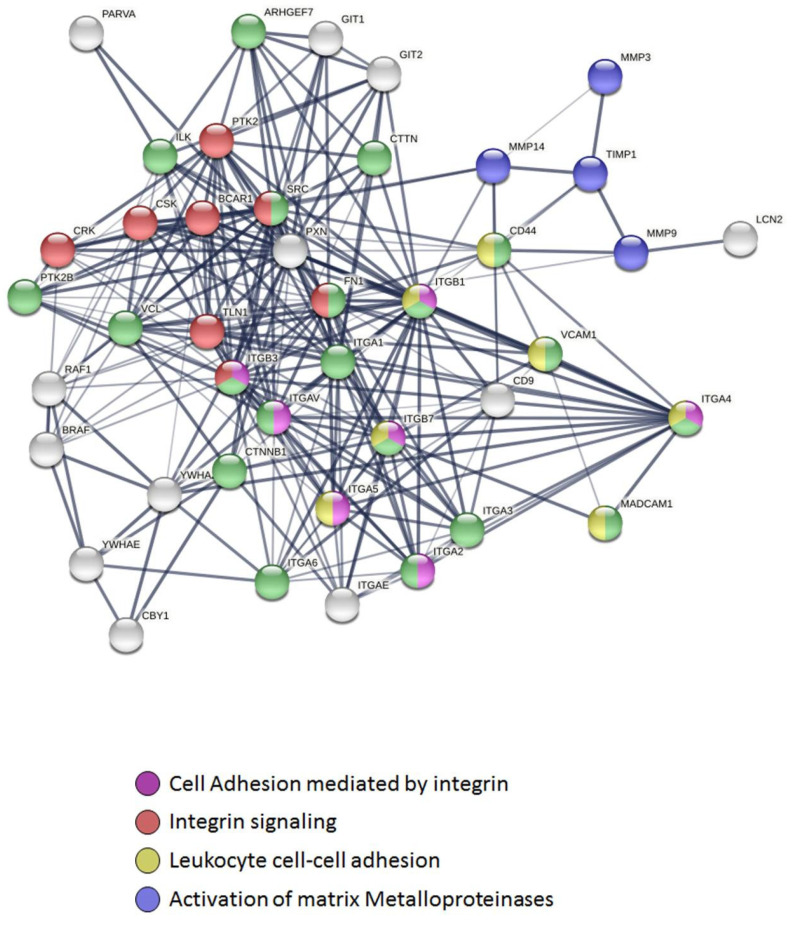
Effect of the *Tn*P peptide on the interaction with Homo sapiens ITA4 using the STRING database. Blockade of ITA4/VCAM-1 complex by *Tn*P generates compromise posterior signaling through MMPs, as well as TIMP-1 and adhesion molecules including other integrins and vinculin (VCL), PTK2B, and integrin-mediated signaling molecules.

**Figure 5 pharmaceuticals-15-00994-f005:**
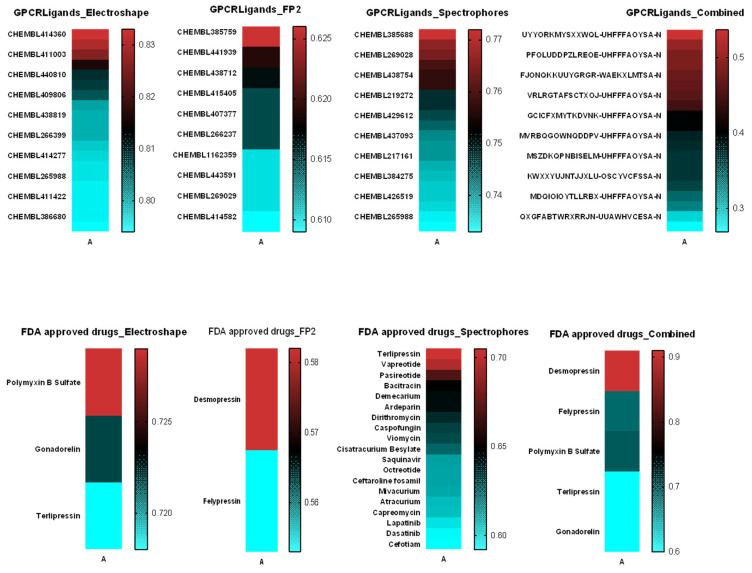
Similarity of *Tn*P. We applied SWISS-MODEL homology modeling tool enhanced by ChEMBL databases and ElectroShape virtual screening method that combines shape and electrostatic information to identify potential drug candidates that can interact with the protein of interest.

**Figure 6 pharmaceuticals-15-00994-f006:**
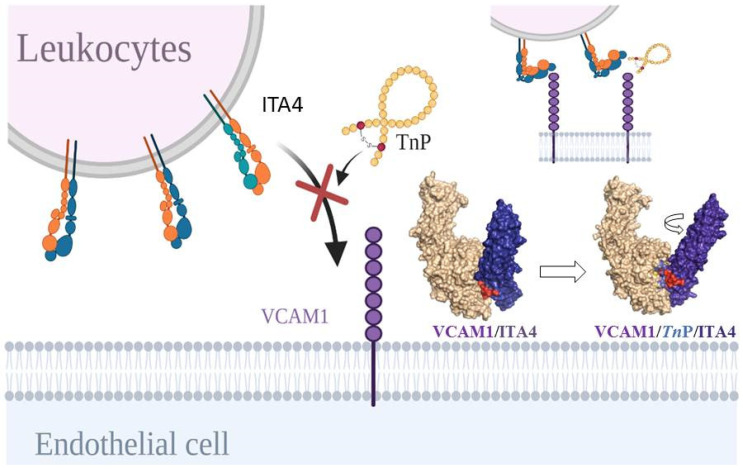
The cartoon illustrates the possible mechanism of action of *Tn*P. The in silico model built to predict the possible molecular targets of *Tn*P showed a harmonic resonance between the anti-inflammatory effects and the computational modeling. The binding mode of *Tn*P showed that it overlapped the ITA4/VCAM-1 binding cleft, leading to a decrease in interaction, with a subsequent reduction in the binding strength and protein fixation in the bending form that decreases its activation. The PDB file ITA4 and VCAM-1 were acquired from SWISS-MODEL using the reference ITA4_HUMAN and VCAM-1_HUMAN.

## Data Availability

Data is contained within the article.

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
