# Peer review of "Shedding Light on the Drug–Target Prediction of the Anti-Inflammatory Peptide TnP with Bioinformatics Tools"

_pharmaceuticals, 2022, doi:10.3390/ph15080994_

Round 1

Reviewer 1 Report

Dear Authors

The article describes tnp family of synthetic cyclic peptides, which is in pre-clinical stage of developmental studies for chronic inflammatory diseases such as asthma and multiple sclerosis. In an experimental autoimmune encephalomyelitis model, the authors found that tnp controls neuroinflammation and prevents demyelination due to its capacity to cross the blood-brain and to act in the central nervous system. The results show that α4β1 integrin is an important protein and is considered responsible to regulate tnp governed pharmacological effects.

The article is well written and understandable. English language is good. The conclusions section is OK. Figures are readable and with good quality. I could not find problems with the methods used.

Therefore, I would recommend the article to be published as it is.

Author Response

MDPI

Pharmaceuticals

Section: Medicinal Chemistry

Special Issue: Drug Candidates from Venoms

Article: 1793018R1

Shedding light on drug-target interaction by bioinformatics tools: Integrin a4β1 is a Possible Target of TnP

Carla LIMA, Silas Fernandes ETO, Monica LOPES-FERREIRA

Reply to Reviewer 1

Thanks for reading the manuscript and providing interesting comments as feedback, we appreciate that.

Open Review

English language and style

( ) Extensive editing of English language and style required
( ) Moderate English changes required
(x) English language and style are fine/minor spell check required
( ) I don't feel qualified to judge about the English language and style

Yes

Can be improved

Must be improved

Not applicable

Does the introduction provide sufficient background and include all relevant references?

(x)

( )

( )

( )

Are all the cited references relevant to the research?

(x)

( )

( )

( )

Is the research design appropriate?

(x)

( )

( )

( )

Are the methods adequately described?

(x)

( )

( )

( )

Are the results clearly presented?

(x)

( )

( )

( )

Are the conclusions supported by the results?

(x)

( )

( )

( )

Comments and Suggestions for Authors

Dear Authors

The article describes tnp family of synthetic cyclic peptides, which is in pre-clinical stage of developmental studies for chronic inflammatory diseases such as asthma and multiple sclerosis. In an experimental autoimmune encephalomyelitis model, the authors found that tnp controls neuroinflammation and prevents demyelination due to its capacity to cross the blood-brain and to act in the central nervous system. The results show that α4β1 integrin is an important protein and is considered responsible to regulate tnp governed pharmacological effects.

The article is well written and understandable. English language is good. The conclusions section is OK. Figures are readable and with good quality. I could not find problems with the methods used.

Therefore, I would recommend the article to be published as it is.

Reviewer 2 Report

The paper submitted for publication in Pharmaceuticals entitled “Shedding light on drug-target interaction by bioinformatics tools: Integrin a4β1 is a Possible Target of TnP” by Carla Lima, Silas Fernandes Eto and Monica Lopes-Ferreira basically result in the the fact that α4β1 integrin is identified as an important key protein, specially because it is considered responsible for regulating the pharmacological effects governed by tnp. This computational study will help to understand how tnp induces its anti-inflammatory effects and might facilitate the identification of possible side effects. It tries to show a link with multiple biologically important targets in humans.

The journal abbreviations are wrong in some cases or not used in other cases. In addition, this section is a bit poor, and not inclusive enough.

There are some typos such as “that TnP could cross-reacts

The study is conducted in a proper way. One concern is that the simplification that Integrin a4β1 is so special is not that clearly demonstrated. The authors should reinforce why, and with which margin of error with respect to other ones is better. Overall, if the authors are able to address those changes I will recommend it for publication in Pharmaceuticals.

Author Response

MDPI

Pharmaceuticals

Section: Medicinal Chemistry

Special Issue: Drug Candidates from Venoms

Article: 1793018R1

Shedding light on drug-target interaction by bioinformatics tools: Integrin a4β1 is a Possible Target of TnP

Carla LIMA, Silas Fernandes ETO, Monica LOPES-FERREIRA

Reply to Reviewer 2
First of all, we appreciate you taking the time out to share your comments with us. We value and respect your opinion. We have already adjusted the manuscript, including additional changes as suggested.

Open Review

English language and style

( ) Extensive editing of English language and style required
( ) Moderate English changes required
(x) English language and style are fine/minor spell check required
( ) I don't feel qualified to judge about the English language and style

Yes

Can be improved

Must be improved

Not applicable

Does the introduction provide sufficient background and include all relevant references?

(x)

( )

( )

( )

Are all the cited references relevant to the research?

( )

(x)

( )

( )

Is the research design appropriate?

(x)

( )

( )

( )

Are the methods adequately described?

(x)

( )

( )

( )

Are the results clearly presented?

(x)

( )

( )

( )

Are the conclusions supported by the results?

( )

(x)

( )

( )

Comments and Suggestions for Authors

The paper submitted for publication in Pharmaceuticals entitled “Shedding light on drug-target interaction by bioinformatics tools: Integrin a4β1 is a Possible Target of TnP” by Carla Lima, Silas Fernandes Eto and Monica Lopes-Ferreira basically result in the the fact that α4β1 integrin is identified as an important key protein, specially because it is considered responsible for regulating the pharmacological effects governed by tnp. This computational study will help to understand how tnp induces its anti-inflammatory effects and might facilitate the identification of possible side effects. It tries to show a link with multiple biologically important targets in humans.

The journal abbreviations are wrong in some cases or not used in other cases. In addition, this section is a bit poor, and not inclusive enough.

There are some typos such as “that TnP could cross-reacts”

The study is conducted in a proper way. One concern is that the simplification that Integrin a4β1 is so special is not that clearly demonstrated. The authors should reinforce why, and with which margin of error with respect to other ones is better. Overall, if the authors are able to address those changes I will recommend it for publication in Pharmaceuticals.
